# High-Resolution and Large-Sensing-Range Liquid-Level Sensor Based on Optical Frequency Domain Reflectometry and No-Core Fiber

**DOI:** 10.3390/s22124480

**Published:** 2022-06-14

**Authors:** Guolu Yin, Pengxi Yang, Hu Xiao, Yu Wang, Zeheng Zhang, Fabing Yan, Tao Zhu

**Affiliations:** 1Key Laboratory of Optoelectronic Technology & Systems (Ministry of Education), Chongqing University, Chongqing 400044, China; yangpengxi2022@163.com (P.Y.); xiaoqhun@163.com (H.X.); m17883657287@163.com (Y.W.); zzh724610754@163.com (Z.Z.); 20180801034@cqu.edu.cn (F.Y.); zhutao@cqu.edu.cn (T.Z.); 2State Key Laboratory of Coal Mine Disaster Dynamics and Control, Chongqing University, Chongqing 400044, China

**Keywords:** liquid level, fiber sensor, optical frequency domain reflectometry, no-core fiber, distributed sensing

## Abstract

Liquid-level sensors are required in modern industrial and medical fields. Optical liquid-level sensors can solve the safety problems of traditional electrical sensors, which have attracted extensive attention in both academia and industry. We propose a distributed liquid-level sensor based on optical frequency domain reflectometry and with no-core fiber. The sensing mechanism uses optical frequency domain reflectometry to capture the strong reflection of the evanescent field of the no-core fiber at the liquid–air interface. The experimental results show that the proposed method can achieve a high resolution of 0.1 mm, stability of ±15 μm, a relatively large measurement range of 175 mm, and a high signal-to-noise ratio of 30 dB. The sensing length can be extended to 1.25 m with a weakened signal-to-noise ratio of 10 dB. The proposed method has broad development prospects in the field of intelligent industry and extreme environments.

## 1. Introduction

Liquid-level sensing is widely used in the fields of modern industry and medical equipment for purposes such as monitoring oil storage, flood warning, sewage treatment plants, chemical process monitoring, and pharmaceutical development [1]. In general, the necessity of measurement resolution and dynamic range for liquid-level sensors cannot be overemphasized. In recent years, ultrasonic [2], electronic [3,4], and mechanical [5] liquid-level sensors have been diffusely applied with the useful features of large dynamic measurement ranges and high sensing resolution. However, their application is limited if the liquid being monitored is conductive or corrosive or when the environment is potentially explosive. Optical fiber liquid-level sensors stand out for their inherent advantages, such as miniature size, non-metallic characteristics, anti-electromagnetic interference, remote sensing capability, and resistance to corrosion, which makes them more suitable for applications where explosive, corrosive, conductive, and flammable hydrocarbons are present.

Generally speaking, liquid-level sensors based on fiber gratings and interference effects can achieve a high sensing resolution, but the sensing range is small. Examples of such sensors include multimode interferometers [6,7,8,9,10], Mach–Zehnder interferometers [11,12,13], anti-resonance reflecting optical waveguides [14,15], Fabry–Perot interferometers [16], fiber Bragg gratings [17,18], and long-period fiber gratings [19,20]. In 2011, Antonio-Lopez et al. reported a no-core fiber (NCF) liquid-level sensor based on the self-image effect of multimode interference [6]. In 2015, M. Sun et al. proposed and demonstrated a new liquid-level sensor based on an in-fiber Mach–Zehnder interferometer, which was formed by cascading two peanut-shaped fiber structures in single-mode fiber [11]. In 2016, H. Chang et al. realized an ultra-sensitive liquid-level sensor using etched chirped fiber Bragg grating, but the measurement range was only 5.6 µm [17]. In 2019, D. Liu et al. designed a high-sensitivity, hollow-core fiber structure liquid-level sensor based on anti-resonance reflecting optical waveguides, which achieved a high-precision liquid-level measurement of 7 × 10^−4^ mm with a measurement range of 4.7 mm [14]. In 2020, J. Wang et al. proposed a liquid-level sensor based on an optical reflective microstructure fiber probe, presenting an ultra-high resolution of 10 µm, but the linear sensing range was only 0.5 mm [12].

In order to expand the measurement range, various distributed optical fiber sensing technologies have been successively applied to the field of liquid-level measurement in recent years. Such technologies include optical frequency domain reflectometry (OFDR) [21], Brillouin optical time-domain analysis (BOTDA) [22], chaos optical correlation domain reflectometry (COCDR) [23], and phase-sensitive optical time-domain reflectometry (φ-OTDR) [24]. In 2018, Christian M. Petrie et al. used the combination of a heating wire and OFDR to achieve liquid-level measurement with a resolution of 5 mm under a 220 mm measurement range by measuring the temperature of the spatial distribution [21]. In 2020, based on the different thermal diffusion rates of liquid and air, H. Zhang et al. realized a distributed liquid-level sensor with a range of 20 cm and a resolution of 1 cm by combining a self-heating high-attenuation fiber and BOTDA [22]. Although the sensing distance was improved in combination with a distributed approach, the measurement resolution was reduced. On the other hand, it is understood that almost all current distributed liquid-level measurements are indirect measurements based on the thermo-optic effect. Since the thermal conductivity of liquid and air are quite different, one can ascertain the level of the interface by means of sensing the vertical distribution of the temperature field. However, the mentioned method usually has to utilize a metal medium to achieve an active heating purpose, which sacrifices the intrinsic safety advantage of silica-based optical fibers.

In this paper, we propose a distributed liquid-level measurement method based on the strong reflection of the evanescent field at the liquid–air interface, which abandons the traditional method based on the different thermal conductivities of liquid and air. The principle of the proposed method has been fully discussed, and its feasibility is also demonstrated experimentally via an OFDR system and an NCF. The experimental results reveal that the proposed system can simultaneously achieve high precision, long sensing range, as well as a high signal-to-noise ratio (SNR) liquid-level measurement for oily substances.

## 2. Operational Principles and Experimental Design

### 2.1. Principles

Figure 1 shows the structure diagram of OFDR. The linear sweeping light from the tunable laser is separated into the main and auxiliary interferometers by using a 1:99 coupler. The 1% light is introduced into an auxiliary interferometer to compensate for the nonlinear scanning effect caused by the laser source. The 99% light is future split by another 1:99 coupler to the main Mach–Zehnder interferometer with a local arm and a measurement arm. The measurement arm of the main interferometer is composed of fiber under test (FUT), and then the reflected Rayleigh scattering light interferes with the reference arm at the balanced photodetector, which is collected and processed by the acquisition card and computer.

Ideally, the laser is completely linearly swept, and the optical frequency emitted by the laser in one cycle can be expressed as:(1)$$f(t)=f_{0}+\gamma t$$
where *f*_0_ is the initial optical frequency of the laser and *γ* is the frequency-sweeping speed of the laser. Compared with the reference signal, the reflection at a certain position in the measuring arm will produce a delay. The delay time has the following relationship with the reflection position:(2)$$\tau_{z}=2n_{g}z/c$$
where *τ_z_* is the delay time, *z* is the fiber distance, *c* is the propagation speed of light in a vacuum, and *n_g_* denotes the effective refractive index of the optical fiber. Considering the laser’s sweeping speed and the time delay, the beat frequency between signals in the reference and measurement arms can be written as:(3)$$f_{b}=\gamma \tau_{z}=2n_{g}\gamma z/c$$

Therefore, the reflection signals in the FUT are directly linked to the beat signal. In data processing, the reflection information of the sensing fiber in the distance domain can be obtained by a Fourier transform of the signal collected by the acquisition card. Due to the existence of the nonlinear sweep effect of the laser, a non-uniform beat signal is generated. If the Fourier transform is directly performed on the original signal, it is expected to be able to observe the diffusion of the bandwidth and the energy in the distance domain, and hence the spatial resolution of the OFDR system is eventually degraded [25]. In order to address this problem, an auxiliary interferometer is constructed in the OFDR system, and the instantaneous laser frequency can be obtained by the Hilbert transform. Then, the beat signal in the main interferometer is resampled at equal frequency intervals in the frequency domain [26]. In this way, the nonlinear sweep effect of the laser is effectively suppressed so as to ensure the high spatial resolution in the distance domain. The theoretical spatial resolution is determined by the laser’s sweeping range:(4)$$\Delta Z=\frac{\lambda^{2}}{2n_{g}\Delta \lambda }$$
where Δ*Z* is the spatial resolution, *λ* is the central wavelength, and Δ*λ* is the scanning range of the laser wavelength.

Figure 2 shows the proposed sensor structure for liquid-level sensing. An NCF with its coating removed is spliced with a lead-in SMF, which is directly used as the liquid-level sensing fiber. The NCF can be regarded as a multimode fiber with air cladding. When the NCF is immersed in the oil liquid, the light will quickly leak into the oil due to the higher refractive index of the oil than that of the silicon dioxide. At the oil–air interface, the evanescent field is reflected back and coupled back to the NCF as the back-propagating modes. When NCF is inserted into the oil solution, the light is guided out, and the Fresnel reflection occurs at the oil–air interface:(5)$$R={\left(\frac{n_{1}-n_{2}}{n_{1}+n_{2}}\right)}^{2}\;$$
where *n*_1_ is the refractive index of air, *n*_2_ is the refractive index of oil, and *R* is the Fresnel reflection coefficient. Since the Fresnel reflected light intensity is much greater than Rayleigh scattering, a strong reflection spike appears in the range domain. In order to capture the strong reflection at the oil–air interface, we employed an OFDR system to measure the distributed reflection along the SMF and the NCF. As the liquid level changes, the reflection peak drifts in the distance domain, and the liquid level can be accurately ascertained according to the position of the reflection peak.

### 2.2. Simulation Analysis

We simulated the light transmission in the NCF by using the beam propagation method. The simulated SMF-NCF structure has an SMF length of 3 mm and an NCF length of 27 mm. the NCF is immersed in glycerol at 11 mm. Figure 3 shows the light propagation and intensity evolution. It can be seen that the power of light at the SMF-NCF fusion point drops sharply, and more energy is coupled to the higher-order modes in the NCF. It is clearly found that the energy starts to leak into the glycerol at the air–liquid interface. At present, the beam propagation simulation cannot reveal the interface reflection and the reverse coupling, but the strong reflection at the air–liquid interface can be observed later in our experiment. It is interesting to find some irregular energy bumps in the NCF section, which are induced by the multi-modes inference and can be further confirmed by the reflection signals in the distance domain by our experimental results.

### 2.3. Experimental Design

Figure 4 shows the experimental setup used for liquid-level sensing. The homemade OFDR system [27] was connected with the liquid-level sensor. An NCF with a diameter of 125 µm was mounted on the electric lifting platform with a step resolution of 1 µm, and the electric lifting platform enabled the optical fiber to be slowly inserted into the oil. The sensing fiber is tensioned to be straightened and suspended from the glass plate so as to vertically immerse it into the oil. Since the refractive indices of oils are very similar, and they are basically slightly larger than the refractive index of the optical fiber (n(SiO_2_) = 1.449), this paper uses stable and easily accessible glycerin (n(glycerin) = 1.473) for the experiment. In these experiments, the tunable light source is a Luna laser (Phoenix 1400) with a wavelength sweeping range from 1515 to 1565 nm. The laser scanning speed is set to 100 nm/s, the sampling rate of the DAQ is 80 MHz, the acquisition time is 0.4096 s, and the corresponding laser sweep range is 40.96 nm. Hence, the spatial resolution of the OFDR system is 20 μm according to Equation (4). Since the OFDR system has the characteristics of high spatial resolution and real-time monitoring, the liquid level can be reflected in the distance domain of OFDR, so the OFDR system can be used to achieve high-precision, real-time monitoring of the liquid level.

## 3. Experimental Results

Figure 5 shows the relative reflection intensity in the SMF and NCF. Because of the mismatch of mode field diameters between SMF and NCF, a strong loss of 15 dB is observed at the splice point between SMF and NCF. Due to the strong loss characteristics of the NCF, the emitted light is basically lost, and the received reflected light is very weak. Hence, compared with a smooth Rayleigh scattering in the SMF, a much-jagged scattering curve was observed in the NCF section. This difference may come from the mode interference in the NCF while only a single mode is propagating in SMF. The scattering signal in NCF has a relatively low level, even approaching the noise floor. It is worth pointing out that the liquid-level measurement does not depend on the scattering signal in the NCF but rather on the strong reflection signal at the oil–air interface. At this location, the forward-propagating evanescent field quickly leaks into the oil and is reflected at the oil–air interface, which is then coupled back to the NCF. Therefore, a much stronger reflection peak obviously appears at the oil–air interface. The scatter level immediately drops to the background noise from behind the oil–air interface. The SNR was defined as the ratio between the strong reflection peak and the noise floor, and some specific SNRs reach as high as 30 dB. The liquid level can be easily determined according to the strong reflection peak with a high SNR.

Figure 6 shows the drift of the reflectance peaks with the change of liquid level at every 3 mm distance in the liquid level range from 53 to 74 mm. As shown in Figure 5, some spurious peaks will appear at specific positions in the distance domain each time. It is interesting to find that these spurious peaks are stable in the air and located at fixed positions. Once the liquid level reaches the position of spurious peaks, these spurious peaks will be submerged in the strong reflected signal. Due to the stability of the spurious peaks, these spurious peaks can be effectively removed by subtracting the initial scattering signal from the measured scattering signal. Figure 6a is the distance domain drift before removing spurious peaks, and Figure 6b is the distance domain drift after removing spurious peaks. It can be seen that the reflection peaks in the distance domain also show uniform changes with the uniform change of the liquid level.

In order to analyze the dependence of the liquid-level sensor on the refractive index, we carried out liquid-level measurement experiments under different refractive indexes. A total of 8 groups of refractive index solutions were prepared by mixing the glycerin with water. The refractive index of the mixed solution was measured with an Abbe refractometer. The result is shown in Figure 7, the proposed liquid-level sensor works only when the ambient refractive index is higher than that of silica. Once this condition is met, it has almost the same sensing performance under different refractive indexes. In the practical application of oil level measurement, the refractive index of oil is almost higher than that of silica. Therefore, the proposed liquid-level sensor has practical application value and can be applied to different oil level monitorings.

In order to explore the advantages of the sensor with a large measurement range and high resolution, this work designed two sets of experiments on the same optical fiber. As shown in Figure 5, the electric control of the electric lift enables the optical fiber to be slowly and vertically inserted into the liquid containing glycerol with a step distance of 5 mm and a total distance of 175 mm.

Figure 8a shows the reflection peak shift for the 175 mm sensing range with a 5 mm step, and the reflection peaks are normalized and smoothed. It can be seen that the reflection peak is very distinct and changes uniformly with the increase in the liquid level. The peak value of the reflection peak on the distance domain with an interval of 5 mm is extracted as the liquid level (measuring distance), and the measurement position and the absolute position (step distance controlled by the electric lift) are linearly fitted. The result is shown in Figure 8b, the slope of the fitting curve is 0.9739, and the R^2^ value is 99.92%, indicating that there is a high degree of linear relationship between the measured position and the absolute position. It is verified that the sensor has a wide range of liquid-level measurements up to 175 mm.

Considering the attenuation of the NCF, the large reflection peak at the oil–air interface is expected to be reduced when extending the NCF sensing length. Therefore, we fabricated another sensing sample with an NCF length of 1.25 m. Figure 9 shows the reflection intensity of the sensing probe in the distance domain. The titled curve reveals the large attenuation of the NCF. Compared with the sensing length of 175 mm, the reflected signal at the oil–air interface is significantly reduced but still has an SNR of 10 dB. When the liquid level is increased by 10 cm, the reflected signal at the end moves 10 cm to a short distance and maintains an SNR of 10 dB. Such an SNR is good enough to show the liquid level position. In conclusion, we obtained a sensing length of 175 mm with an SNR of 30 dB and a sensing length of 1.25 m with an SNR of 10 dB.

In order to verify that the sensor has the advantages of high resolution at the same time, the following experiments are designed on the same fiber. As shown in Figure 10, the electric control of the electric lift enables the optical fiber to be inserted into the liquid containing glycerin with a step distance of 0.1 mm, with a total distance of 2 mm. Figure 10a shows that the distance domain reflectance peak of OFDR changes uniformly with the increase in the liquid level. In order to better reflect the change process, the distance domain reflection peaks in Figure 10a are obtained after normalization and smoothing.

The peak value of the reflection peak on the distance domain is extracted as the liquid level (measurement distance), measurement position, and absolute position (step distance controlled by the electric lift) are linearly fitted. The result is shown in Figure 10b: the slope of the fitting curve is 0.9905, and the R^2^ value is 99.89%, indicating that there is a highly linear relationship between the measured position and the absolute position.

To verify the stability of the sensor, a repeated experiment was designed at the same liquid level. Figure 11a shows several typical scattering signals in the distance domain. Figure 11b shows 120 groups of liquid-level data points, collected every 30 s for a total of 60 min. It is found that all scattering curves almost overlap together, and the strong reflection peaks fluctuate within a very tiny range. Slight changes may occur over time due to ambient noise and glycerin viscosity. These results indicate that the proposed liquid-level sensor has a good stability of ±15 μm.

## 4. Discussion

The current resolution of 0.1 mm may be limited by the multi-modes in NCF. The evanescent fields of all modes are superimposed together at the liquid–air interface. The respective modes have different effective refractive indices, and the reflective peaks are not located at the same positions in the distance domain but have a slight position difference. Therefore, the overlapping of all reflections makes the reflection peak broaden. We can further improve the resolution by optimizing the sensor to reduce the modes in the evanescent field. In addition, the viscous nature of the measured glycerol may also have an effect on the accuracy of the sensor, resulting in a reduction in the resolution of the sensor.

The sensing distance of the proposed liquid-level sensor depends on the attenuation of the NCF. The SNR of the reflected signal at the oil–air interface can reach up to 30 dB when the sensing length is 175 mm, and the sensing length can be extended to 1.25 m by sacrificing the SNR. The SNR of 10 dB with a sensing length of 1.25 m is still good enough to show the liquid-level position.

Table 1 shows the performance comparison between the distributed liquid-level sensor proposed in this paper and various reported optical fiber liquid-level sensors. Compared with other liquid-level sensors, this liquid-level sensor has the characteristics of a large measuring range and a high resolution. NCF has been usually spliced between two SMFs for working as the liquid-level sensors based on the principle of multi-mode interference [6,7,8,9,10]. Because part of the NCF was immersed in the liquid and part was exposed to the air, the change in liquid level induced wavelength shifts in the interference spectrum. However, this kind of liquid-level sensor, based on measuring wavelength shift, is easily affected by temperature. In order to eliminate the influence of temperature, fiber Bragg gratings were chosen to compensate for the temperature effect [9]. In this paper, the principle of the liquid-level sensor is based on the high reflected peak in the NCF instead of on the wavelength shift. The position of the reflected peak position at the oil–air interface depends on the refractive index and the optical fiber length according to Equation (2). For the silica, the thermal optic coefficient and thermal expansion coefficient are 6.1 × 10^−6^ °C and 0.55 × 10^−6^ °C, respectively. The change in the distance information caused by the change in the refractive index and thermal expansion is so small as to be neglected. Therefore, this sensor also has the advantage of insensitivity to temperature and can be used in harsh environments.

Compared with the method based on φ-OTDR in 2021 [24], our proposed method achieves almost the same sensing resolution and sensing range. However, the high resolution of 0.142 mm in the φ-OTDR method was realized by the approach of densely winding optical fibers. It is worth noting that almost all previous distributed liquid-level measurements are indirect measurements through temperature measurements according to the different thermal conductivities of liquid and air. In this paper, the sensing mechanism of these indirect measurements is abandoned, and the direct measurement of the liquid level is realized by using OFDR to demodulate the evanescent field strength reflection peak of the NCF at the liquid–air interface. This competitive performance makes this liquid-level sensor intrinsically safe without heating in the modern industrial and medical fields.

## 5. Conclusions

This paper proposes a simple and novel oil level sensor based on OFDR and NCF. The sensor achieves 0.1 mm high-precision measurement and is expected to achieve ultra-high-precision measurement with OFDR theoretical spatial resolution of 20 µm; the measurement range reaches 175 mm with an SNR of 30 dB. When the sensing length is increased to 1.25 m, the SNR is reduced to 10 dB due to the attenuation of the NCF. Because this sensor uses OFDR to interrogate the distance domain information on the optical fiber, the change in temperature to the length of the optical fiber is almost negligible. In summary, the proposed sensor can achieve high-precision, wide-range, high-SNR, and temperature-insensitive liquid-level measurement of oily substances. The proposed intrinsically safe liquid-level sensor is expected to find applications in the modern industrial and medical fields.

## Figures and Tables

**Figure 1 sensors-22-04480-f001:**
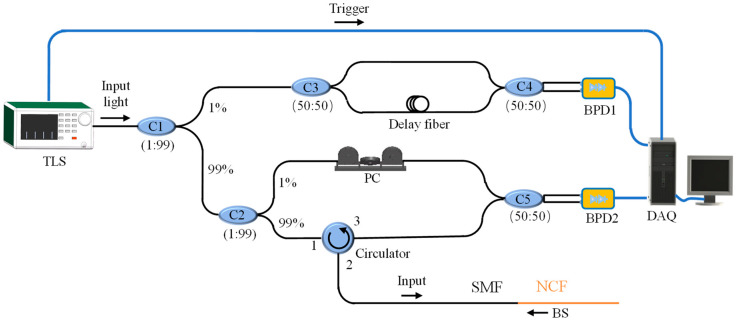
The system schematic of the optical frequency domain reflectometry. TLS: tunable laser source; C1-5: couplers; BPD1-2: balanced photodetectors; PC: polarization controller; DAQ: data acquisition; SMF: single-mode fiber; NCF: no-core fiber; BS: back scattering.

**Figure 2 sensors-22-04480-f002:**
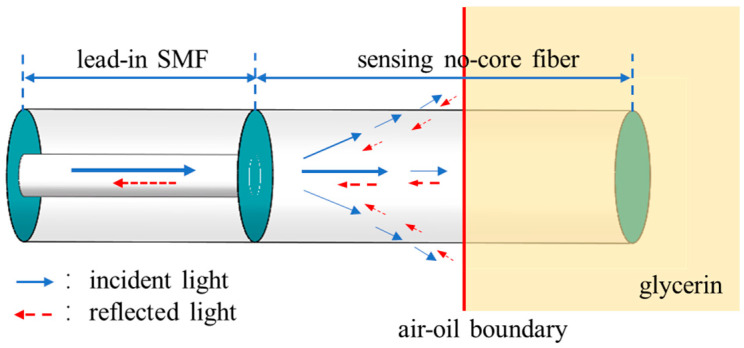
Schematic of the operation mechanism of a liquid-level sensor.

**Figure 3 sensors-22-04480-f003:**
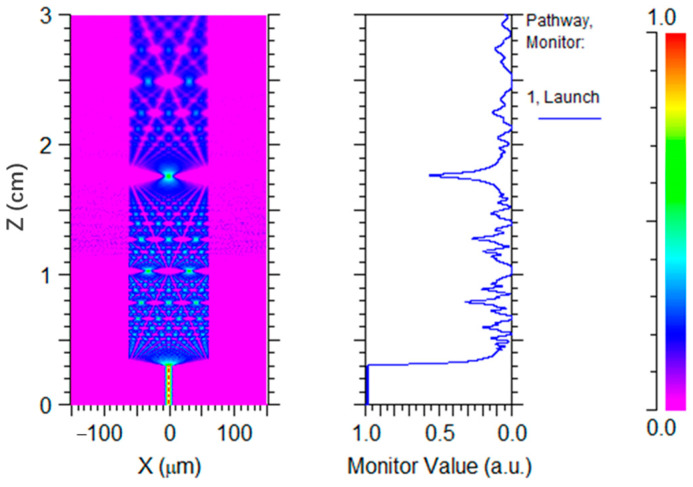
Simulation analysis of the light propagation in an SMF-NCF structure.

**Figure 4 sensors-22-04480-f004:**
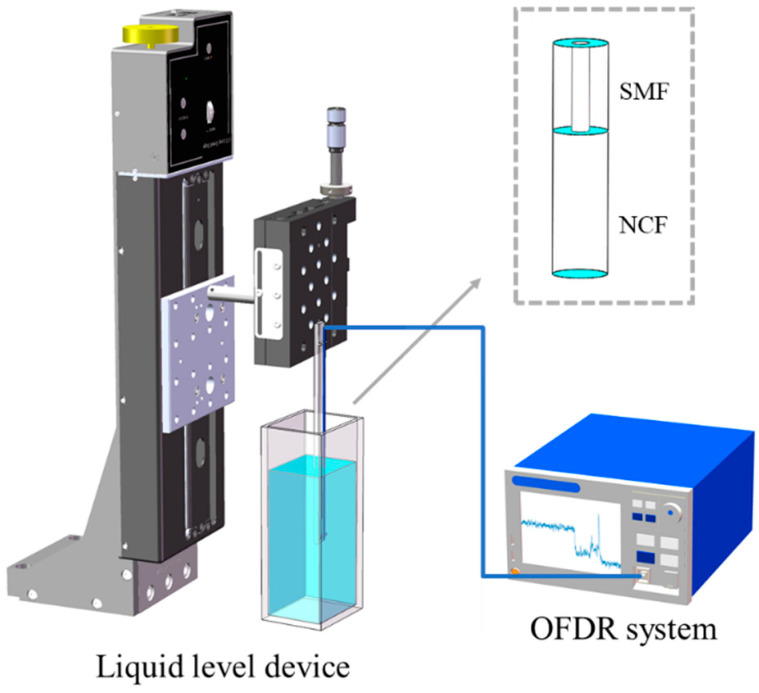
Experimental setup for liquid-level measurement. OFDR: optical frequency domain reflectometry; NCF: no-core fiber; SMF: single-mode fiber.

**Figure 5 sensors-22-04480-f005:**
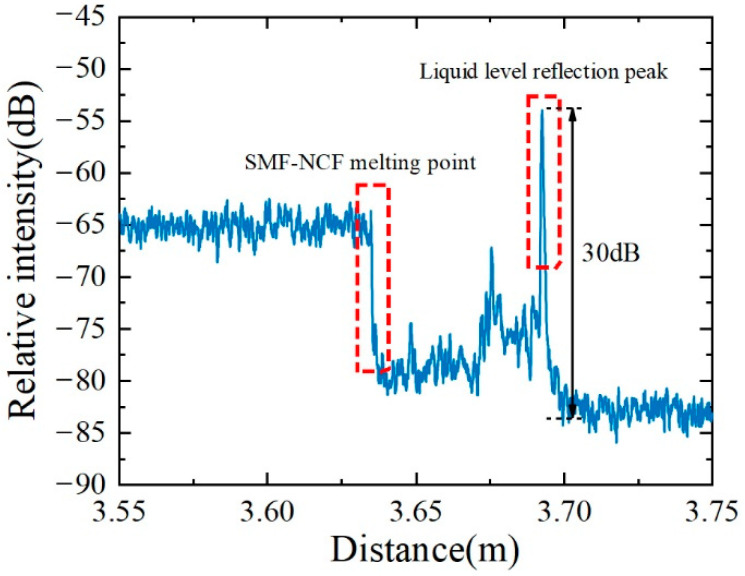
Typical reflection intensity of the liquid-level sensor in the distance domain.

**Figure 6 sensors-22-04480-f006:**
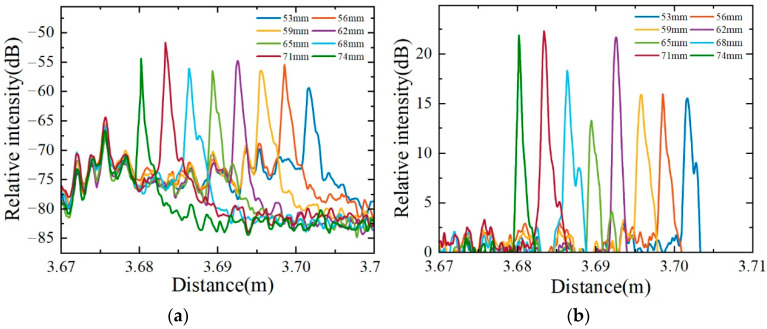
Reflection peak drift with liquid level from 53 to 74 mm. (**a**) Original reflection peaks in the distance domain; (**b**) distance domain drift after removing spurious peaks.

**Figure 7 sensors-22-04480-f007:**
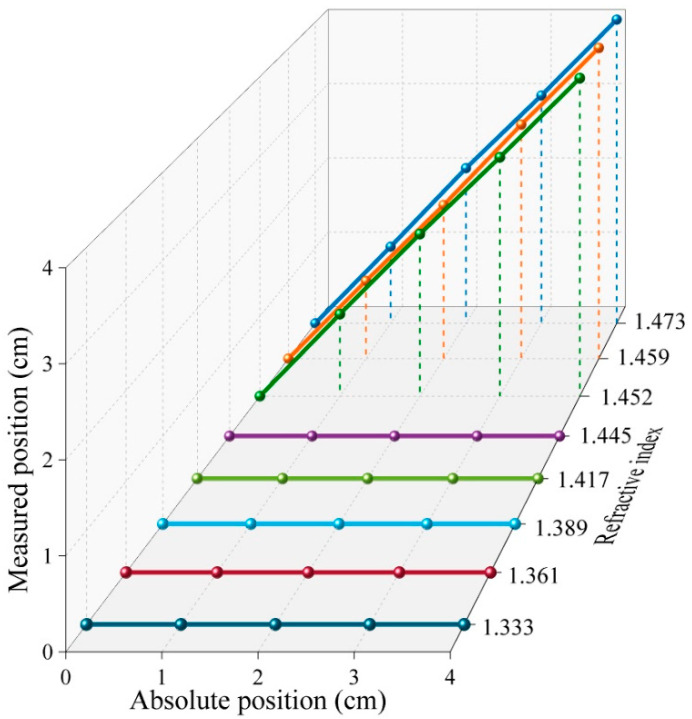
The liquid-level sensor under different refractive indexes from 1.333 to 1.473.

**Figure 8 sensors-22-04480-f008:**
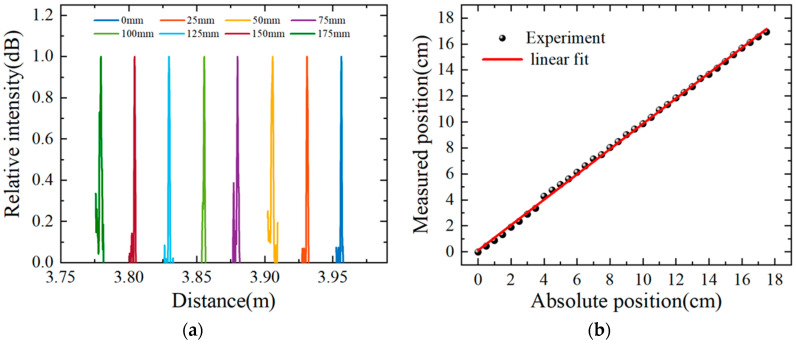
Liquid-level sensor in a range of 175 mm with an interval of 5 mm. (**a**) Normalized reflection peak drift with liquid level; (**b**) linear fit of measured position and absolute position of liquid level.

**Figure 9 sensors-22-04480-f009:**
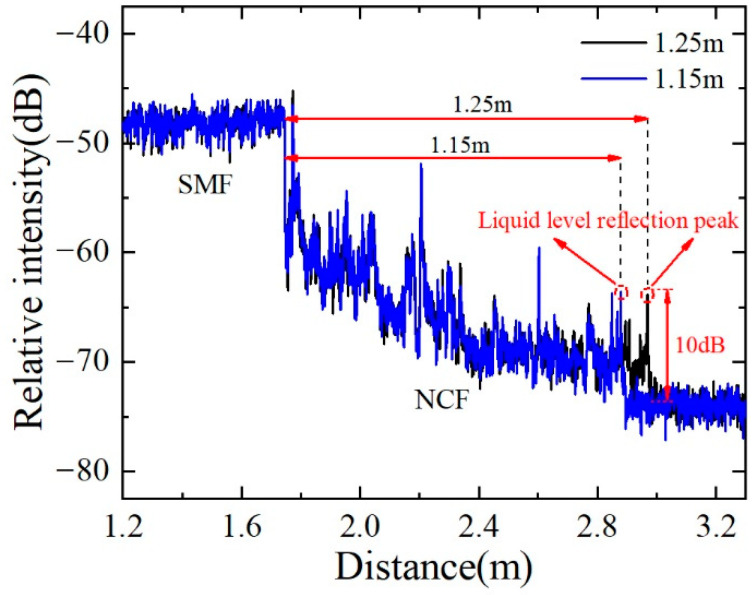
Reflection intensity of the liquid-level sensor with a sensing length of 1.25 m.

**Figure 10 sensors-22-04480-f010:**
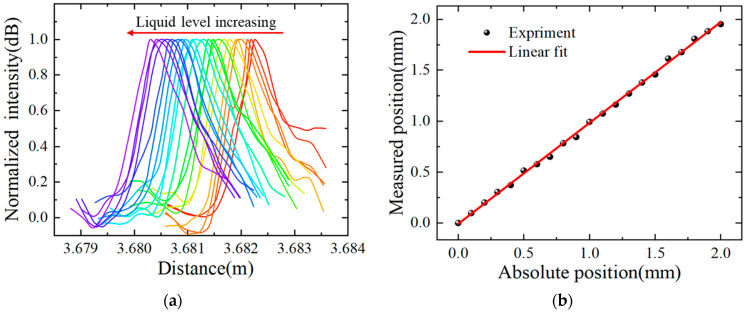
Liquid-level sensor in a range of 2 mm with an interval of 0.1 mm. (**a**) Normalized reflection peak drift with liquid level; (**b**) linear fit of measured position and absolute position of liquid level.

**Figure 11 sensors-22-04480-f011:**
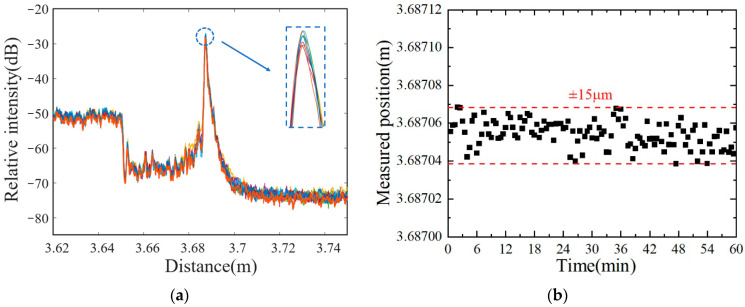
Stability experiment of the liquid-level sensor. (**a**) Typical reflection signals in the distance domain; (**b**) reflection positions collected every 30 s for 60 min.

**Table 1 sensors-22-04480-t001:** Sensing resolution and sensing range of several liquid-level sensors.

Method	Resolution (mm)	Sensing Range (mm)
Hollow core fiber [14]	7 × 10^−4^	4.7
NCF and FBG [8]	0.46	45
Reflective microfiber probe [12]	0.01	0.5
SMF-NCF-SMF [9]	0.5	500
OFDR and heater wire [21]	5	220
BOTDA and self-heated high attenuation fiber [22]	10	200
φ-OTDR and thermal optic effect with cylinder [24]	0.142	200
This work	0.1	12,500

## Data Availability

Not applicable.

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
