# Peer review of "High-Resolution and Large-Sensing-Range Liquid-Level Sensor Based on Optical Frequency Domain Reflectometry and No-Core Fiber"

_sensors, 2022, doi:10.3390/s22124480_

Round 1
Reviewer 1 Report
The paper is very interesting and may find many potential readers.
The theory and results are well presented.
However, before publication, I recommend making some improvements to the text:
1. In Fig. 4(a), the colors of the curve and corresponding line for 74 m in the legend are different. The curve's color is navy blue, and the bar is red.
2. In Fig 6(a), it is pretty hard to interpret this picture. There are too many curves, and the colors for some curves are the same. A better presentation of these results would improve the readability of these results.
3. Page 7, line 208: I think the expression "wide-range" would be better than "large-range."
After addressing these concerns, I think the paper can be published in the "Sensors."
Author Response
We provided a point-by-point response to the reviewer’s comments, Please see the attachment

Reviewer 2 Report
This paper presents a liquid/oil level sensor based on an OFDR system and no-core fiber as sensing fiber spliced with SMF. Several concerns need to be addressed before the final acceptance of this manuscript. Moreover, there are many grammatical, and English errors throughout the manuscript, which need revisions before the final manuscript submission for publication. Some of the major concerns are;
- Due to high splice loss between SMF and NCF, and high attenuation of NCF, the sensing range cannot be longer. The authors failed to measure the dynamic range and cannot see the difference between the scattering signal from NCF and the noise floor as shown in Figure 3.
- It is well investigated in the literature, that when RI changes the reflection from the evanescent field of the no-core fiber at the liquid-air interface. Simply using NCF spliced to SMF is not novel.
- The operating principles need to be further emphasised with proper equations.
- What are the reasons for non-uniform scattering from the NCF section?. The spurious peaks will appear at specific positions in the distance domain each time. How many measurements were taken to remove the spurious peaks, as shown in Figure 4 (b)? The stability of peak reflection over some time (at least 30 min.) needs to be investigated.
- There is no evidence of temperature-insensitive liquid level measurement in the proposed method. Have you investigated temperature effects on your measurements by changing temperatures of glycerin liquid?.
Author Response
We provided a point-by-point response to the reviewer’s comments. Please see the attachment

Round 2
Reviewer 2 Report
The authors addressed all comments and concerns very well. The manuscript technically improved. I suggest doing English proofreading before the final submission.